# Authentication of Argan (*Argania spinosa* L.) Oil Using Novel DNA-Based Approaches: Detection of Olive and Soybean Oils as Potential Adulterants

**DOI:** 10.3390/foods11162498

**Published:** 2022-08-18

**Authors:** Joana S. Amaral, Fatima Z. Raja, Joana Costa, Liliana Grazina, Caterina Villa, Zoubida Charrouf, Isabel Mafra

**Affiliations:** 1REQUIMTE-LAQV, Faculdade de Farmácia, Universidade do Porto, Rua de Jorge Viterbo Ferreira, 228, 4050-313 Porto, Portugal; 2Centro de Investigação de Montanha (CIMO), Instituto Politécnico de Bragança, Campus de Santa Apolónia, 5300-253 Bragança, Portugal; 3Laboratório Associado para a Sustentabilidade e Tecnologia em Regiões de Montanha (SusTEC), Instituto Politécnico de Bragança, Campus de Santa Apolónia, 5300-253 Bragança, Portugal; 4Faculty of Science, University Mohammed V, Avenue Ibn Battouta, Rabat 10106, Morocco

**Keywords:** argan oil, authenticity, adulterant detection, real-time PCR, quantification, *Olea europaea*, *Glycine max*

## Abstract

Argan oil is a traditional product obtained from the fruits of the argan tree (*Argania spinosa* L.), which is endemic only to Morocco. It is commercialized worldwide as cosmetic and food-grade argan oil, attaining very high prices in the international market. Therefore, argan oil is very prone to adulteration with cheaper vegetable oils. The present work aims at developing novel real-time PCR approaches to detect olive and soybean oils as potential adulterants, as well as ascertain the presence of argan oil. The ITS region, *matK* and *lectin* genes were the targeted markers, allowing to detect argan, olive and soybean DNA down to 0.01 pg, 0.1 pg and 3.2 pg, respectively, with real-time PCR. Moreover, to propose practical quantitative methods, two calibrant models were developed using the normalized ΔCq method to estimate potential adulterations of argan oil with olive or soybean oils. The results allowed for the detection and quantification of olive and soybean oils within 50–1% and 25–1%, respectively, both in argan oil. Both approaches provided acceptable performance parameters and accurate determinations, as proven by their applicability to blind mixtures. Herein, new qualitative and quantitative PCR assays are proposed for the first time as reliable and high-throughput tools to authenticate and valorize argan oil.

## 1. Introduction

Argan (*Argania spinosa* L.) is a slow-growing tree endemic only in Morocco. To protect the unique argan forest in southwestern Morocco, in 1998, the UNESCO declared the Arganeraie (the argan tree and its ecological system) as a biosphere reserve. Later, in 2014, the “practices and know-how concerning the argan tree” were inscribed in the UNESCO’s Representative List of the Intangible Cultural Heritage of Humanity. The most emblematic use of this tree regards the production of argan oil, a cold-pressed non-refined vegetable oil. Argan oil is traditionally obtained from the fruits of the argan tree using a laborious multistep process that includes fruit picking, fruit peeling, nut cracking, kernel roasting, kernel grinding, dough malaxing and oil collection [1,2,3]. Despite the current use of modern mechanical presses for oil extraction to allow for a higher oil yield, the process still results in a very low production (approximately 4 L of oil per 100 kg of dried argan fruit), requiring laborious work corresponding to 20 person-hours [4]. Argan oil is produced in different grades, namely, for food and cosmetic purposes [5]. Edible argan oil, registered as a product with the Protected Geographical Indication (PGI) since 2011, is obtained from lightly roasted kernels conferring a hazelnut flavor to the oil, while argan oil used for cosmetics is obtained from raw kernels. In recent decades, numerous studies have shown the nutritional and dermo-cosmetic benefits of this oil [2,6,7], which have been known for centuries and transmitted among generations of Berbere women. Due to its properties and successes as an ingredient in cosmetic products, currently, argan oil is considered one of the most prized oils in the world, with a growing worldwide demand [6]. As a premium product, argan oil is highly prone to adulteration by partial or even total substitution with other vegetable oils. Therefore, different methodologies have been proposed for argan oil authentication, mostly relying on chemical markers analyzed with chromatographic approaches [8]. Hilali et al. [9] proposed the use of campesterol, a sterol present in argan oil in very low amounts (<0.4%), as an adulteration marker allowing for the detection of 2% additions of campesterol-rich vegetable oils and 5% of olive, apricot and hazelnut oils, which are naturally low in campesterol. The use of tocopherols was not so successful, as it allowed for the detection of adulterant oils only above 5% [10]. Ourrach et al. [11] suggested the combined use of 3,5-stigmastadiene, chlorophyllic pigments and hydrocarbon fractions to detect up to 5% additions of refined olive and sunflower oils and virgin olive oil. The same level of detection was achieved based on the triacylglycerol (TAG) profile determined with liquid chromatography coupled to evaporative light-scattering [12] and photodiode-array [13] detection. In addition, spectroscopic approaches [14,15,16,17] and, more recently, selected-ion flow-tube mass spectrometry (SIFT-MS) spectra [18] have also advanced as fast screening tools for argan oil authentication. Despite their rapidity and minimal sample preparation requirements, these methods involve the use of chemometrics to predict the level of oil adulteration, which demands very large numbers of samples to construct proper databases towards robust mathematical models.

Considering that several factors, such as edaphoclimatic conditions, development stage, plant part and age, among others, are known to affect the plant’s chemical composition [19], DNA molecules have emerged as alternative and unambiguous markers for plant species identification in vegetable oils, which are independent from those factors. Particularly, DNA-based techniques have been successfully used for detecting genetically modified organisms (GMO) in refined oils [20,21,22] and for the authentication of vegetable oils, such as olive oil [23] and several refined vegetable oils [24,25]. Real-time polymerase chain reaction (PCR) combined with high-resolution melting (HRM) analysis was used by Vietina et al. [26] to detect the addition of maize and sunflower to olive oil down to a limit of 10% and by Ganapoulos et al. [27] to detect 1% of canola oil admixed with olive oil. Moreover, DNA-based methods have demonstrated their feasibility in the identification of plant species in complex matrices [28,29,30,31,32,33,34]. Therefore, in this work, novel approaches based on DNA markers are proposed for the first time to detect argan oil and the presence of soybean and olive oils as its potential adulterants.

## 2. Materials and Methods

### 2.1. Sampling and Reference Oil Mixtures

Fresh leaves and nuts of *Argania spinosa* L. were directly collected from trees in the region of Agadir, Morocco, while leaves of *Olea europaea* L. were collected in the region of Viseu, Portugal. Additionally, other plant species also used in the production of oil were tested in cross-reactivity studies, including walnut, sunflower, maize, almond, hazelnut, cashew nut, pistachio nut, peanut, Brazil nut, macadamia nut, pine nut, rapeseed, oat and rye (Appendix A). The fresh leaves and nuts were oven dried at 30 °C in the dark and were ground in a Grindomix GM200 laboratory mill (Retsch, Haan, Germany).

Authentic argan oil was kindly supplied by the Groupement des Coopératives Targanine (Agadir, Morocco). Commercial argan oil samples of food and cosmetic grades were acquired in Morocco (Appendix A). Samples of extra-virgin olive oil and refined soybean oil were acquired from local supermarkets in Porto, Portugal. Binary model mixtures were prepared by adding well-known quantities of olive oil to argan oil in the proportions of 50, 25, 10, 5 and 1% (*w/w*), and adding known amounts of soybean oil to argan oil in the proportions of 40, 25, 10, 5 and 1% (*w/w*). Additionally, two sets of binary mixtures containing 7.5% and 15% (*w/w*) of olive oil or soybean oil in argan oil were prepared for method validation.

### 2.2. DNA Extraction

Before DNA extraction, the oil mixtures were centrifuged as suggested by Costa et al. [20,21]. For that purpose, 300 g of oil was weighed into six centrifuge tubes, which were centrifuged at 18,514× *g* for 30 min at 4 °C. The supernatant was discarded until half of each tube was left, and the remaining oil/residue was centrifuged for another 30 min in the same conditions. The supernatant of each tube was carefully removed through pipetting and the residual pellets were collected, combined in one tube and then centrifuged for 30 min (18,514× *g*, 4 °C). The residual pellet was transferred to one 2 mL sterile reaction tube, centrifuged for 30 min in the same conditions and the supernatant was discarded. Afterwards, the residual pellet was submitted to DNA extraction using protocol B of the Nucleospin^®^ Plant II (Macherey-Nagel, Düren, Germany) kit. Briefly, 300 µL of buffer PL2 pre-heated at 65 °C was added to each tube and incubated at 65 °C for 1 h with continuous mixing (1000 rpm) and occasional vortex mixing. After incubation, the procedure was followed according to the manufacturer’s instructions. DNA extracts were stored at −20 °C until analysis.

### 2.3. DNA Quality and Purity

A UV spectrophotometer, using a Synergy HT multi-mode micro-plate reader (BioTek Instruments, Inc., Winooski, VT, USA) and the Take3 micro-volume plate accessory, was used to assess the yield and purity of DNA extracts. The nucleic acid quantification protocol for dsDNA samples in the Gen5 data analysis software version 2.01 (BioTek Instruments, Inc., Winooski, VT, USA) was used to determine the DNA content. The ratio of the absorbance at 260 and 280 nm (A260/A280) was determined as the purity parameter of the extracted DNA.

Electrophoresis in 1% agarose gel stained with 1× Gel Red (Biotium, Hayward, CA, USA) and ran in 1× SGTB buffer (GRISP, Porto, Portugal) for 20–25 min at 200 V was performed to evaluate the integrity of the DNA extracts. Agarose gels were visualized under a UV light tray Gel Doc™ EZ System (Bio-Rad Laboratories, Hercules, CA, USA) and a digital image was acquired with Image Lab software version 5.2.1 (Bio-Rad Laboratories, Hercules, CA, USA).

### 2.4. Oligonucleotide Primers

For the specific identification of olive and argan DNA, sequences of the chloroplastidial *matK* gene of *O. europaea* L. and the nuclear region of the internal transcribed spacer 2 (ITS2) of *A. spinosa* L. were retrieved from the NCBI database (http://www.ncbi.nlm.nih.gov/ accessed on 8 November 2021) (accession numbers AJ429335.1 and AM408056.1, respectively). Primers were designed using the Primer-BLAST software tool (http://www.ncbi.nlm.nih.gov/tools/primer-blast/, accessed on 8 November 2021) (Table 1). Primer specificity was assessed in silico using the same tool and the basic local alignment search tool BLAST (http://blast.ncbi.nlm.nih.gov/Blast.cgi, accessed on 8 November 2021). Primer properties and the absence of self-hybridization and hairpins were verified using the software OligoCalc (http://www.basic.northwestern.edu./biotools/oligocalc.html, accessed on 8 November 2021).

For soybean detection, primers targeting the lectin gene [35] were previously designed, as well as universal eukaryotic primers targeting the conserved nuclear 18S rRNA gene, used to assess the amplification capacity of the DNA extracts [36] (Table 1).

The oligonucleotide primers used in this work were synthesized by Eurofins Genomics (Ebersberg, Germany).

### 2.5. Qualitative PCR

PCR amplifications were carried out in a total reaction volume of 25 μL, containing 2 μL of DNA extract (10 ng), 67 mM Tris-HCl (pH 8.8), 16 mM of (NH_4_)_2_SO_4_, 0.1% of Tween 20, 200 µM of each dNTP, 1.0 U of SuperHot Taq DNA Polymerase (Genaxxon Bioscience GmbH, Ulm, Germany), 2.0 mM of MgCl_2_, 200 nM (ITS2A-F/ITS2A-R, matKO-F/matKO-R) or 280 nM (LE1/LE2, EG-F/EG-R) of each primer (Table 1). The reactions were performed in a MJ Mini™ Gradient Thermal Cycler (Bio-Rad Laboratories, Hercules, CA, USA) using the following programs: (i) initial denaturation at 95 °C for 5 min; (ii) 35 cycles at 95 °C for 30 s, 62 °C (ITS2A-F/ITS2A-R, matKO-F/matKO-R) or 60 °C (LE1/LE2) or 63 °C (EG-F/EG-R) for 30 s and 72 °C for 30 s; (iii) final extension at 72 °C for 5 min. Each extract was amplified at least in duplicate assays.

Electrophoresis was carried out in a 1.5% agarose gel containing Gel Red 1× (Biotium, Hayward, CA, USA) for staining and SGTB 1× (GRiSP, Research Solutions, Porto, Portugal) was used to confirm amplicons. Agarose gels were visualized under a UV light tray Gel Doc™ EZ System (Bio-Rad Laboratories, Hercules, CA, USA) and a digital image was obtained with Image Lab software version 5.2.1 (Bio-Rad Laboratories, Hercules, CA, USA).

### 2.6. Real-Time PCR

Real-time PCR amplifications were carried out in 20 μL of total reaction volume, containing 2 µL of DNA (10 ng to 0.01 pg), 1× SsoFast EvaGreen Supermix (Bio-Rad Laboratories, Hercules, CA, USA), 300 nM (ITS2A-F/ITS2A-R, matKO-F/matKO-R) or 350 nM of (LE1/LE2) each primer (Table 1). A fluorometric thermal cycler CFX96 Real-time PCR Detection System (Bio-Rad Laboratories, Hercules, CA, USA) was used with the following conditions: 95 °C for 5 min, 45 cycles at 95 °C for 10 s and 65 °C for 40 s, with the collection of the fluorescence signal at the end of each cycle. The data evaluation from each real-time PCR assay was performed using the software Bio-Rad CFX Manager 3.1 (Bio-Rad Laboratories, Hercules, CA, USA). Real-time PCR assays were performed, at least, in two independent runs using *n* = 4 replicates each time.

Calibration curves were constructed using 10-fold serially diluted DNA extracts (10 ng–0.01 pg), which allowed for determining the absolute limits of detection (LOD) and quantification (LOQ). The acceptance criteria for real-time PCR assays were established according to the MIQE guidelines (minimum information for publication of quantitative real-time PCR experiments) [37] and the definition of minimum performance requirements for analytical methods of GMO testing [38]. Accordingly, the following parameters, namely, the slope between −3.6 and −3.1, the PCR efficiency within 90–110% and the correlation coefficient (*R*^2^) ≥ 0.98 were established [37,38]. The sensitivity was expressed as the LOD, which was the lowest amount or concentration that could be reliably detected (the lowest amplified level for 95% of the replicates). The LOQ was the lowest amount or concentration of analyte in a sample that could be reliably quantified with an acceptable level of trueness and precision, which was determined as the lowest amplified level within the linear dynamic range of the calibration curve. The dynamic range should cover a minimum of 4 orders of magnitude and, ideally, extend to 5 or 6 log10 concentrations [37,38].

## 3. Results

### 3.1. DNA Quality and Selection of Target Region

For method development and optimization, DNA was successfully extracted from argan and olive leaves, as well as soybean flour, achieving suitable yields within 7.9–16.9 ng/µL, 5.8–8.4 ng/µL and 49.0–99.7 ng/µL, with purities (A260/A280) of 1.4–2.2, 1.7–-2.0 and 1.8–2.0, respectively. The DNA yields for the oil mixtures and commercial argan oil samples were in the range of 5.2–9.3 ng/µL, with purities of 1.5–2.1. Despite the low DNA yields, all the extracts showed a suitable amplification capacity as inferred from the strong PCR fragments targeting a universal and conserved gene (18S rRNA) (Appendix A).

To specifically detect argan and olive DNA, new primer sets were designed targeting the ITS and *matK* regions, respectively, because both are recognized barcode markers with high species discriminatory powers. The ITS is a robust phylogenetic marker at the species level, while *matK* has a high evolutionary rate and suitable length [39]. Moreover, both markers may provide highly sensitive methods because *matK* is a chloroplastidial gene that is present in multiple copies and the ITS is a nuclear region present in multiple ribosomes of nuclear DNA. The results of the PCR optimization for the detection of *A. spinosa* and *O. europaea* confirmed the high sensitivity of the assays, achieving 1 pg and 0.1 pg, respectively (Figure 1A,B). The specificity of the assays was firstly assessed with an in silico analysis and further confirmed experimentally using several non-target species that are commonly used in food and cosmetic oils (Appendix A). The detection of soybean was carried out by targeting the lectin gene using previously designed primers (LE1/LE2) [35] to amplify a short PCR amplicon (103 bp). The choice was justified by their successful application to detect soybean DNA in refined oils [20,21]. As shown in Figure 1C, the soybean-specific PCR detection was down to 0.8 ng of DNA. The three specific PCR assays were then applied to the commercial samples of argan oil, confirming the presence of argan in all samples (Appendix A) and the absence of olive and soybean DNA as potential adulterants (Appendix A).

### 3.2. Real-Time PCR Assays

#### 3.2.1. Absolute LOD and LOQ

After selecting the species-specific markers, real-time PCR assays using EvaGreen dye were successfully developed for each target species (Figure 2). Figure 2A,C,E show the amplification curves and respective derivative melting curves that provided the single melt peaks for each species at 89.43 ± 0.14 °C, 77.43 ± 0.10 °C and 78.79 ± 0.04 °C, supporting the specificity of the target amplification and the absence of non-specific amplicons. The calibration curves obtained for *A. spinosa*, *O. europaea* and *G. max* (Figure 2B,D,F) showed that all performance parameters, namely, the PCR efficiency (102.0 to 104.2%), slope (−3.276 to −3.226) and *R*^2^ (0.995 to 0.998), complied with the acceptance criteria established for real-time PCR assays [37,38]. In addition, the dynamic ranges covered seven, six and four orders of magnitude, achieving limits of detection (LOD) of 0.01 pg, 0.1 pg and 3.2 pg of DNA for argan, olive and soybean, respectively (Figure 2B,D,F). Since the LOD values were within the dynamic range, the limits of quantification (LOQ) could be considered as the same values.

#### 3.2.2. Construction of the Normalized Calibration Curves

To estimate the potential adulterations of argan oil with olive or soybean oils, two quantitative models were developed using the ΔCq method. This approach has been frequently applied to several complex and/or highly processed food matrices [28,29,36]. It is based on the construction of a normalized calibration curve, plotting the difference between the quantification cycle values of the target sequence and a universal reference marker (ΔCq) versus the log of the concentration of the target species. Therefore, this approach can reduce the influence of potential PCR inhibitors, DNA degradation and low DNA yields, which are critical issues when amplifying DNA from oil matrices [20,21]. For the construction of the calibration curves, two sets of binary reference mixtures of olive oil in argan oil (50, 25, 10, 5 and 1% (*w/w*)) and soybean oil in argan oil (40, 25, 10, 5 and 1% (*w/w*)) were prepared as calibrants, and the respective ΔCq values were plotted against the concentration of each target adulterant oil (Figure 3). The obtained calibration curves showed values of PCR efficiency (82.9 and 81.8%) and slopes (−3.8124 and −3.8521) slightly out of the acceptance criteria, but with acceptable correlations (*R*^2^> 0.98), within the ranges of 50–1% and 25–1% for olive/argan and soybean/argan oil mixtures, respectively. However, it is important to refer to that, in the case of samples where it was difficult to extract high-quality DNA, such as oils, a slope within −4.1 and −3.1 and a PCR efficiency of 75–110% were acceptable [38]. Both calibration curves provided LOD and LOQ of 1% of adulterant oil in argan oil, being able to estimate olive oil until 50% (Figure 3A) and soybean oil until 25% (Figure 3B), because above this value, the PCR efficiency and correlation values were not acceptable. However, a dynamic range of 25–1% could be considered feasible for the purpose of estimating eventual adulterations.

#### 3.2.3. Validation of Quantitative Real-Time PCR Systems

To assess the performance of the two quantitative normalized PCR systems, two sets of blinded samples containing 7.5% and 15% of adulterant (olive or soybean) in argan oil were used. The estimation of the respective oil contents allowed for assessing the performance of the assays regarding trueness and precision (Table 2). The coefficients of variation expressed the relative standard deviations of results and were obtained under repeatability conditions, showing acceptable values that were within 0.41% and 13.1% (≤25%) and attesting to the precision of both systems. The measured trueness was expressed as the bias, whose values ranged between −24% and 22.9%, being within ±25% of the actual values, which confirmed the closeness of agreement between the tested and the actual values of both systems [38].

## 4. Discussion

In recent years, there has been rising interest towards the authentication of foods, including vegetable oils [40] and botanicals [31], using molecular makers. DNA molecules are ubiquitously present in all cells, resistant to harsh conditions, such as food processing, and independent from plant age and tissue, climatic, geographical or agronomical factors. Therefore, DNA markers have been considered unequivocal identifiers for the traceability and authentication of food, with several advantages over proteins that are less resistant to processing and chemical markers, which might vary with edaphoclimatic conditions. In relation to their application to vegetable oils, several advances have been actualized, mainly regarding olive oil authentication [23,26,27,40,41] and the detection of GMO in soybean oil [20,21,40]. However, none of the reports addressed the authentication of argan oil, and most of them lacked any quantitative analyses, being mainly confined to the quantification of GMO in soybean oil [20,21].

In the present work, the suitability of using DNA markers was exploited for the authentication of argan oil for the first time. For that purpose, unequivocal markers were identified for the detection of argan oil and two potential adulterants, namely, olive and soybean oils. The target markers were the ITS, *matk* and *lectin* genes, providing the detection of argan, olive and soybean DNA down to 0.01 pg, 0.1 pg and 3.2 pg, respectively, with real-time PCR. The three species-specific assays provided calibration curves that complied with the acceptance criteria concerning PCR efficiency, slope and *R*^2^ values (Figure 2). Additionally, to propose two practical quantitative methods to estimate the potential adulterations of argan oil with olive or soybean oils, two calibrant models were developed using the ΔCq method (Figure 3). The feasibility of this approach was already demonstrated in processed food matrices, such as species authentication in meat products [42], herbal products [29] and spices [36], and the detection and quantification of potentially allergenic ingredients, namely, soybean [28], lupine [43] and milk [44]. To our knowledge, the application of a normalized ΔCq approach to authenticate vegetable oil was herein described for the first time. The two calibrant models allowed for detecting and quantifying olive oil in the range of 50–1% (Figure 3A) in argan oil and soybean oil within 25–1% in argan oil (Figure 3B). Both approaches provided acceptable performance parameters, with proven applicability to blind mixtures and precise and accurate quantitative analyses. In summary, two novel real-time PCR approaches were proposed as specific, sensitive and high-throughput tools to authenticate and valorize argan oil.

## Figures and Tables

**Figure 1 foods-11-02498-f001:**
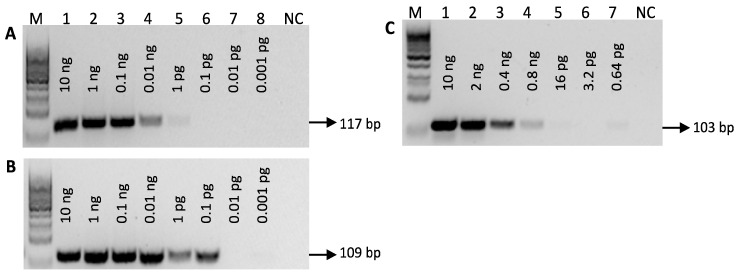
Agarose gel electrophoresis of PCR products targeting ITS, *matK* and *lectin* genes of *A. spinosa* (**A**), *O. europaea* (**B**) and *G. max* (**C**), respectively, using serially diluted DNA of each species. Legend: M, 100 bp molecular marker (Bioron, Ludwigshafen, Germany); lanes 1–8, serially diluted DNA; NC, negative control.

**Figure 2 foods-11-02498-f002:**
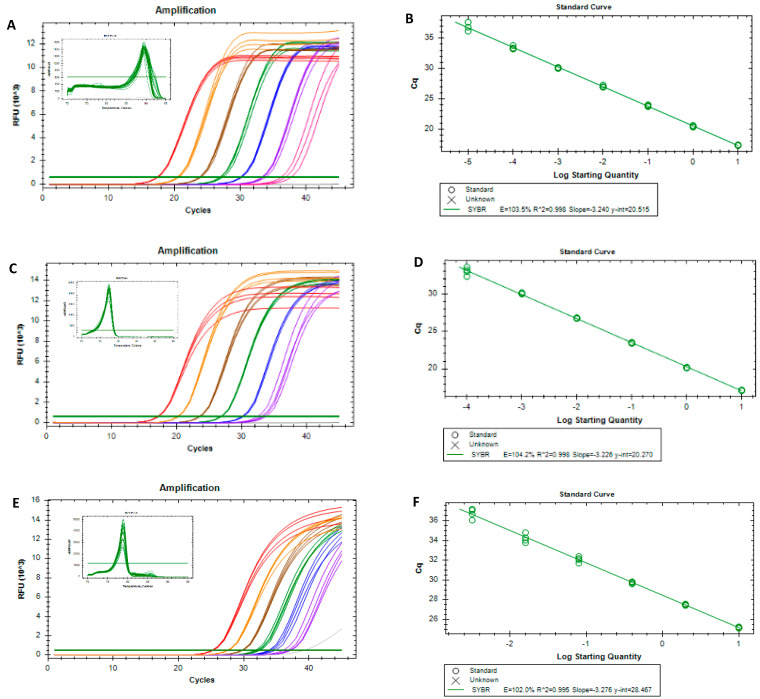
Real-time PCR amplification curves (with respective melting curves) (**A**,**C**,**E**) and calibration curves targeting ITS, *matK* and *lectin* genes of *A. spinosa* (**A**,**B**), *O. europaea* (**C**,**D**) and *G. max* (**E**,**F**), respectively, using 10-fold serially diluted DNA (10 ng to 0.01 pg) for *A. spinosa* and *O. europaea* and 4-fold serially diluted DNA (10 ng to 0.64 pg) for *G. max* (*n* = 4 replicates).

**Figure 3 foods-11-02498-f003:**
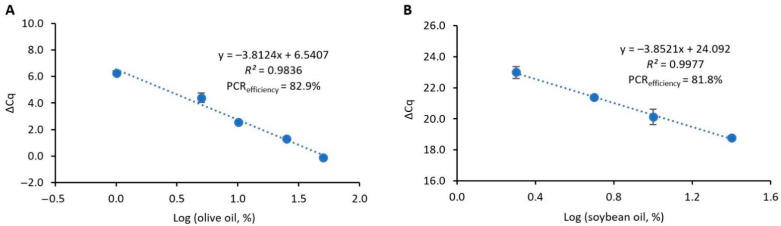
Normalized calibration curves obtained with real-time PCR targeting the ITS region of olive (**A**) and the lectin gene of soybean (**B**), using reference mixtures of olive oil (50, 25, 10, 5 and 1%, *w/w*) in argan oil and soybean oil in argan oil (40, 25, 10, 5 and 1%, *w/w*), respectively (*n* = 8 replicates).

**Table 1 foods-11-02498-t001:** Oligonucleotide primers used in this work.

Species	Target Gene	Primers	Sequence (5′-3′)	Amplicon (bp)	Reference
Argan	ITS2	ITS2A-F	CTCGTCCCGTCCCGCAAAG	117	This work (AM408056.1)
		ITS2A-R	CCACCACTCGTCGTGACGTT		
Olive	*matK*	matKO-F	GCTGTGGTTTCATCCAAGAAGGA	109	This work (AJ429335.1)
		matKO-R	GCTCCGTACCACTGAAGCGT		
Soybean	*Lectin*	LE1	CAAAGCAATGGCTACTTCAAAG	103	[35]
		LE2	TGAGTTTGCCTTGCTGGTCAGT		
Eukaryotic	18S rRNA	EG-F	TCGATGGTAGGATAGTGGCCTACT	109	[36]
		EG-R	TGCTGCCTTCCTTGGATGTGGTA		

**Table 2 foods-11-02498-t002:** Validation results based on the application of the normalized quantitative real-time PCR approaches to blind mixtures containing olive or soybean oils in argan oil.

Samples	Adulterant Oil (%, *w/w*)		SD ^b^	CV (%) ^c^	Bias ^d^
	Actual	Mean Predicted ^a^			
Olive oil in argan oil					
A	7.5	9.7	0.0	0.41	22.9
B	15.0	14.0	0.2	1.1	−7.5
Soybean oil in argan oil					
C	7.5	5.9	0.8	13.1	−21.2
D	15.0	11.4	0.6	5.6	−24.0

^a^ Values are the means of replicate assays (*n* = 6). ^b^ SD, standard deviation. ^c^ CV, coefficient of variation. ^d^ Bias = ((mean value-true value)/true value × 100).

## Data Availability

Data is contained within the article or Appendix A.

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
