# Peer review of "Authentication of Argan (Argania spinosa L.) Oil Using Novel DNA-Based Approaches: Detection of Olive and Soybean Oils as Potential Adulterants"

_foods, 2022, doi:10.3390/foods11162498_

Round 1
Reviewer 1 Report
The manuscript aims at developing novel real-time PCR approaches to detect olive and soybean oils as potential adulterants, as well as ascertain the presence of argan oil. The manuscript is well written and organized. However, there are some places which have to be carefully revised by the authors, before a publication on the journal.
Major comments:
1. Line 67-79. The literature review is not comprehensive enough.
2. For cross-reactivity studies: Table S1 alone is not enough, please provide raw data of agarose gel or real-time PCR.
3. The binary model mixtures are closer to the actual oil samples, why not use the binary mixtures to test the LOD of this method?
4. In section 3.2.3, only two sets of binary mixtures containing 7.5% and 15% (w/w) of olive oil or soybean oil in argan oil were prepared for method validation. It is necessary that low, medium and high levels of olive oil or soybean oil in the binary mixtures are used to verify the accuracy of the method.
5. Commercial argan oil samples were mentioned in the manuscript on line 186, but they were not tested for practicality using the established real-time PCR method. Why?
Minor corrections:
Line 153. Correct “72 °C for 30 s; 72 °C for 30 s;”.
Author Response
Answer to Reviewer 1
The manuscript aims at developing novel real-time PCR approaches to detect olive and soybean oils as potential adulterants, as well as ascertain the presence of argan oil. The manuscript is well written and organized. However, there are some places which have to be carefully revised by the authors, before a publication on the journal.
Major comments:
- Line 67-79. The literature review is not comprehensive enough.
Answer to Reviewer 1 – Three additional references were added (references 22, 24 and 25) and the sentences were changed to improve clarity.
- For cross-reactivity studies: Table S1 alone is not enough, please provide raw data of agarose gel or real-time PCR.
Answer to Reviewer 1 – As suggested, new figures (Fig. S1) of agarose gel electrophoresis were provided to demonstrate the absence of cross-reactivity.
- The binary model mixtures are closer to the actual oil samples, why not use the binary mixtures to test the LOD of this method?
Answer to Reviewer 1 –The levels of the binary reference mixtures for the development of the calibration model are normally prepared to cover the widest dynamic range that is suitable for the objective of the work, which in this case is the detection/quantification of adulterations. Therefore, the binary mixtures were prepared within the range 1% until 50% because adulteration practices below 5% would not make sense and above around 20% would probably be noted (color, odor, general appearance). In fact, the binary mixtures were prepared to determine both, LOD and LOQ, which were both established as 1% and fully described in the section 3.2.2 (lines 248-259). To clarify this issue, the levels of binary reference mixtures were added in line 257.
- In section 3.2.3, only two sets of binary mixtures containing 7.5% and 15% (w/w) of olive oil or soybean oil in argan oil were prepared for method validation. It is necessary that low, medium and high levels of olive oil or soybean oil in the binary mixtures are used to verify the accuracy of the method.
Answer to Reviewer 1 – We agree with the reviewer that the number of blind mixtures is limited. This was due to limitations in the supply of argan oil from the Moroccan cooperative. However, we considered that the two levels (7.5 and 15%), not included in the calibration curves, are sufficient to validate the calibration model since they were critically chosen within the expected fraudulent practices, somewhere between 5 and 20%.
- Commercial argan oil samples were mentioned in the manuscript on line 186, but they were not tested for practicality using the established real-time PCR method. Why?
Answer to Reviewer 1 – The results of the commercial argan oil samples were now included in Fig. S2, as well as their description in a new Table S2 (Supplementary material). They were all positive for argan detection by qualitative PCR, as expected, but olive or soybean oils were not detected, as shown in Fig. S2. Therefore, no real-time PCR results could be presented for the commercial samples.
Minor corrections:
Line 153. Correct “72 °C for 30 s; 72 °C for 30 s;”.
Answer to Reviewer 1 – The duplication was corrected.

Reviewer 2 Report
Review Report for submission foods-1837924-v1 (Authentication of argan (Argania spinosa L.) oil using novel DNA-based approaches: detection of olive and soybean oils as potential adulterants)
Topic: New approaches based on DNA markers (ITS, matk and lectin genes) are proposed to detect argan oil and the presence of soybean and olive oils as potential adulterants, using real-time PCR. In order to authenticate argan oil, two practical quantitative methods are proposed to estimate the potential adulteration of argan oil with olive or soybean oils, two calibration models were developed using the ∆Cq method.
There is a lot of work involved and the ms. is of potential interest.
The topic falls within the aims and scopes of the Foods journal.
Originality = Fair (confirmatory results for the existing literature landscape). Although, apply a method already developed on other materials, even by the same authors.
Author Response
Answer to Reviewer 2
New approaches based on DNA markers (ITS, matk and lectin genes) are proposed to detect argan oil and the presence of soybean and olive oils as potential adulterants, using real-time PCR. In order to authenticate argan oil, two practical quantitative methods are proposed to estimate the potential adulteration of argan oil with olive or soybean oils, two calibration models were developed using the ∆Cq method.
There is a lot of work involved and the ms. is of potential interest.
The topic falls within the aims and scopes of the Foods journal.
Originality = Fair (confirmatory results for the existing literature landscape). Although, apply a method already developed on other materials, even by the same authors.
Answer to Reviewer 2– We appreciate the valuable comments about the manuscript, though we find the last comment not very clear. If the reviewer means that the work lacks originality, we must emphasize that the use of DNA markers is herein proposed for the first time to detect argan oil and the presence of soybean and olive oils as its potential adulterants (lines 76-86). The novelty is also highlighted along the discussion of results. Two new markers and respective methods were proposed to detect argan and olive, while the detection of soybean was based on a previously identified marker. Additionally, two calibration models for the relative quantification of olive oil in argan oil and soybean oil in argan oil were also developed for the first time in this work. Therefore, the present work provides several innovating issues.

Reviewer 3 Report
The article entitled “Authentication of argan (Argania spinosa L.) oil using novel DNA-based approaches: detection of olive and soybean oils as potential adulterants " presents a quite serious and rigorous study. The authors present work aims to develop new real-time PCR approaches to detect olive and soybean oils as potential adulterants, as well as to determine the presence of argan oil. They have developed two calibrant models using the normalized ΔCq method to estimate possible adulterations of argan oil with olive or soybean oils quantitatively.
However, the authors need to clarify and correct the following points:
1.- Page 1, line 22: This is the first time that in the text it appears confused in terms of accuracy. According to ISO, accuracy is the combination of trueness and precision. Therefore, either they use only accuracy or they use trueness and precision.
2.- Page 3, lines 121-125: What criteria or numerical parameters were used to assess integrity?
3.- Page 4, line 166: What excitation wavelength does the equipment use?
4.- Page 4, lines 177-180: Explain better the calculation of LOD and LOQ. The text is not clear.
5.- Page 4, line 184: What do the authors consider suitable content and why?
6.- Page 5-6, lines 223-225: Explain better why they can be considered the same. By definition, the LOD can never be equal to the LOQ.
7.- Page 7, figure 3: Indicate on the x-axis, the % also to improve the reader's comprehension.
8.- Page 7, lines 261 and 266: Again there is confusion about accuracy. Unify criteria.
9.- Page 7, line 261. There is another conceptual confusion. Repeatibility is a way of indicating precision. You cannot indicate: “precision, and repeatability”.
Author Response
Answer to Reviewer 3
The article entitled “Authentication of argan (Argania spinosa L.) oil using novel DNA-based approaches: detection of olive and soybean oils as potential adulterants" presents a quite serious and rigorous study. The authors present work aims to develop new real-time PCR approaches to detect olive and soybean oils as potential adulterants, as well as to determine the presence of argan oil. They have developed two calibrant models using the normalized ΔCq method to estimate possible adulterations of argan oil with olive or soybean oils quantitatively.
However, the authors need to clarify and correct the following points:
1.- Page 1, line 22: This is the first time that in the text it appears confused in terms of accuracy. According to ISO, accuracy is the combination of trueness and precision. Therefore, either they use only accuracy or they use trueness and precision.
Answer to Reviewer 3 – As suggested, we have now used the terminology of ISO 5725, which considers "trueness" and "precision" to describe the accuracy of a measurement method and did the changes accordingly.
2.- Page 3, lines 121-125: What criteria or numerical parameters were used to assess integrity?
Answer to Reviewer 3 – The integrity was assessed through visual inspection of extracted DNA after agarose gel electrophoresis as described in section 2.3. A high molecular weight band means high integrity DNA, while a smeared appearance without any band suggests degraded DNA extracts.
3.- Page 4, line 166: What excitation wavelength does the equipment use?
Answer to Reviewer 3 – In this work, we used the EvaGreen dye as fluorophore, which has similar spectral characteristics to SYBR Green I. Therefore, according to manufacture’s instructions, we choose the most appropriate channel (FAM/SYBR) that has excitation and emission wavelengths of 450-490 nm and 510-530 nm, respectively.
4.- Page 4, lines 177-180: Explain better the calculation of LOD and LOQ. The text is not clear.
Answer to Reviewer 3 – The sentences were changed to improve clarity (lines 178-191). The sensitivity was expressed as the LOD, which is the lowest amount or concentration that can be reliably detected (the lowest amplified level for 95% of the replicates). The LOQ is the lowest amount or concentration of analyte in a sample that can be reliably quantified with an acceptable level of trueness and precision, which was determined as the lowest amplified level within the linear dynamic range of the calibration curve. Such definitions considered the acceptance criteria for real-time PCR assays according to the MIQE Guidelines (Minimum Information for Publication of Quantitative Real-Time PCR Experiments) [37] and the Definition of Minimum Performance Requirements for Analytical Methods of GMO Testing [38] (lines 180-183).
5.- Page 4, line 184: What do the authors consider suitable content and why?
Answer to Reviewer 3 – From our experience, we consider a suitable DNA yield of a minimum concentration of 5 ng/uL because at this level we generally obtain amplifiable DNA, while below this value sometimes no amplifiable DNA is obtained.
6.- Page 5-6, lines 223-225: Explain better why they can be considered the same. By definition, the LOD can never be equal to the LOQ.
Answer to Reviewer 3 – As answered previously, the sensitivity was expressed as the LOD, which is the lowest amount or concentration that can be reliably detected (the lowest amplified level for 95% of the replicates). The LOQ is the lowest amount or concentration of analyte in a sample that can be reliably quantified with an acceptable level of trueness and precision, which was determined as the lowest amplified level within the linear dynamic range of the calibration curve (lines 178-191). Such definitions considered the acceptance criteria for real-time PCR assays according to the MIQE Guidelines (Minimum Information for Publication of Quantitative Real-Time PCR Experiments) [34] and the Definition of Minimum Performance Requirements for Analytical Methods of GMO Testing [35] (lines 180-183).
7.- Page 7, figure 3: Indicate on the x-axis, the % also to improve the reader's comprehension.
Answer to Reviewer 3 – The % was already indicated in both x-axis: Fig. 3A – Log (olive oil, %); Fig. 3B – Log (soybean oil, %).
8.- Page 7, lines 261 and 266: Again there is confusion about accuracy. Unify criteria.
Answer to Reviewer 3 – As previously answered, we have now used the terminology of ISO 5725, which considers "trueness" and "precision" to describe the accuracy of a measurement method and did the changes accordingly.
9.- Page 7, line 261. There is another conceptual confusion. Repeatibility is a way of indicating precision. You cannot indicate: “precision, and repeatability”.
Answer to Reviewer 3 – It was corrected as suggested.

Round 2
Reviewer 1 Report
The manuscript has been successfully revised and now may be published.